

# LPITutor: an LLM based personalized intelligent tutoring system using RAG and prompt engineering

Zhensheng Liu[1,*], Prateek Agrawal[2,3,4,*], Saurabh Singhal[5], Vishu Madaan[2,3], Mohit Kumar[6] and Pawan Kumar Verma[7]

[1] School of Educational Sciences, Bohai University, Jinzhou, Liaoning, China
[2] Computer Science and Engineering, Lovely Professional University, Phagwara, Punjab, India
[3] Faculty of Engineering and Technology, Shree Guru Gobind Singh Tricentenary University, Gurugram, Haryana, India
[4] INTI International University, Nilai, Malaysia
[5] Computer Science and Engineering, Greater Noida Institute of Technology, Greater Noida, Uttar Pradesh, India
[6] Computer Science and Engineering, MIT Art, Design and Technology University, Pune, Maharashtra, India
[7] Computer Science & Engineering, Sharda University, Greater Noida, Uttar Pradesh, India
* These authors contributed equally to this work.

## ABSTRACT

Development of large language models (LLMs) has transformed the landscape of personalized education through intelligent tutoring systems (ITS) which responds to diverse learning requirements. This article proposed a model named LLM based Personalized Intelligent Tutoring System (LPITutor) that is based on LLM for personalized ITS that leverages retrieval-augmented generation (RAG) and advanced prompt engineering techniques to generate customized responses aligned with students' requirements. The aim of LPITutor is to provide customized learning content that adapts to different levels of learners skills and question complexity. The performance of proposed model was evaluated on accuracy, completeness, clarity, difficulty alignment, coherence, and relevance. The finding of LPITutor indicates that it effectively balances the response accuracy and clarity with significant alignment to the difficulty level of student queries. The proposed work also emphasises the broader implications of artificial intelligence (AI)-driven ITS in education and presents future directions for improving the adaptation and optimization of LPITutor.

## INTRODUCTION

Tutoring systems are a significant aspect of education since they offer tailored guidance and support to students. Rule-based algorithms in conventional intelligent tutoring systems (ITS) deliver well-defined learning experiences through providing tailored exercises, feedback, and assessment. Conventional systems are, however, limited by their

Corresponding author
Pawan Kumar Verma,
abes.pawan@gmail.com

reliance on predefined material and a strict domain. Such systems are dependent upon rule-based techniques and pre-stored knowledge graphs to advise students (*Anderson et al., 1995*) and may fail to register each learning requirement of individuals in real time. Due to this reason, these systems tend to be unable to support students' variegated learning requirements and approaches. Secondly, instructors and programmers need to try their level best to author and revise material manually in traditional ITS systems. While this format is helpful in some situations, its inflexibility falls short in managing large and more dynamic educational material, which is increasingly needed in contemporary learning environments (*Modran et al., 2024*; *Nye, Mee & Core, 2023*).

The recent development of large language models (LLMs) has added new dimensions in students' tutoring. LLMs offer significant advances in understanding, generation, and personalization of natural languages and thus make ITS one of its most popular applications. The combination of LLMs with retrieval-augmented generation (RAG) and prompt engineering completely transforms conventional tutoring systems by making educational content more accessible, interactive, and personalized to learners' requirements. With advancement in artificial intelligence (AI) technology and improved educational experiences, the demand for personalized ITS has increased. This more flexible and responsive learning environment customized to the progress and needs of each student makes ITS different from traditional rule-based systems (*Stamper, Xiao & Hou, 2024*).

## Motivation

The limitation of traditional tutoring methods and the growing demand for more personalized, scalable, and flexible learning solutions motivate the development of LLM-based ITS. Its been difficult for educators in a conventional classroom environment to provide individualized attention to every student in large groups. This leads to the gaps in understanding because every student learns at different paces and requires different levels of clarification. ITS have solved this problem, but their effectiveness is limited by their static content and their lack of adaptability to a wide range of students and learning contexts (*Kabir & Lin, 2023*).

The integration of LLMs into tutoring systems with support of RAG and prompt engineering solves these challenges by enabling dynamic content generation and real-time adaptation to each learner's progress. LLMs not only provide immediate and contextually relevant feedback, but also customize their responses to previous interactions of the learner, creating a more personalized and effective learning experience (*Modran et al., 2024*). In addition, LLMs have access to a vast repository of knowledge that enables them to present subject matter in diverse disciplines without the constant manual updates of the educator (*Stamper, Xiao & Hou, 2024*).

Another significant motivation is the growing need for scalable education technologies. Global access to education increases the demand for tools that reach large numbers of students and offer personalized instruction. The LLM-based tutoring system offers the

scalability that enables universal access to high-quality education on a global scale (*Nye, Mee & Core, 2023*).

## Challenges

Although LLM based ITS holds significant potential, both conventional and LLM based tutoring approaches still face several challenges. In traditional teaching environments, an educator faces the primary challenge of providing personalized real-time feedback to students. Educators often struggle to adapt their teaching strategies to meet the needs of the individual learner in a large group of students due to time and resource constraints (*Meyer et al., 2024*). As a result, learners may not receive the targeted support to overcome learning difficulties they require, and thus it creates gaps in their knowledge and understanding.

Conventional ITS offer some level of personalization, but also limit users to their predefined content and rule-based frameworks. These systems often lack the flexibility to cover different courses and to adapt dynamically to a learner's needs. Educators put in significant manual effort to update these systems to include new topics or learning scenarios (*Fu et al., 2023*). This lack of customization and flexibility creates a significant barrier to wider acceptance and limits their effectiveness in supporting a variety of learners across various educational domains.

Although LLMs-based tutoring system addresses these challenges, it also introduces its own set of complexities. One major challenge is to ensure the accuracy and relevancy of LLM-generated content in case of complex educational topics. Sometimes, LLMs generate incorrect responses that could mislead students (*Stamper, Xiao & Hou, 2024*). Other important concerns are related to the ethical use of AI in education, such as data privacy, potential bias in the content created, and the need to ensure fair access to these advanced technologies (*Ilagan, 2023*).

Despite these challenges, LLM-enabled systems with RAG and prompt engineering have the potential to overcome many constraints of traditional tutoring approaches. Educational environments are increasingly interested in implementing LLM based systems because they offer a more scalable, flexible, and personalized approach to learning.

## LPITutor: personalized intelligent tutoring system

The proposed LLM based Personalized Intelligent Tutoring System (LPITutor) model creates a highly personalized ITS. It combines RAG and prompt engineering to offer a unique learning experience that responds to each learner based on the requirements. Unlike conventional tutoring systems, LPITutor does not rely on predefined content; rather, it dynamically generates personalized explanations and learning resources in real time based on the student's progress and interactions. The main contribution of LPITutor is to use RAG to retrieve relevant information from a wide range of external sources and prompt engineering to provide accurate and contextually appropriate responses. The hypothesis for this research is considered as "When RAG, prompt engineering and LLM

are personalized using contextual data specific to the learner, it can generate more accurate, relevant and pedagogically effective responses than traditional LLM-based tutoring systems". To investigate this hypothesis, we formulate the following research questions:

1. How effectively does the LPITutor framework integrate RAG, LLMs, and prompt engineering to generate context-aware and content-based tutoring responses?
2. In what ways does LPITutor personalize instructional content based on learner profiles and how does this personalization impact the quality of learning interactions?
3. How do domain experts and learners perceive the accuracy, relevance, and usefulness of LPITutor-generated responses compared to baseline LLM outputs?
4. What are the ethical, practical, and pedagogical considerations involved in deploying an LLM-based intelligent tutoring system in real-world educational settings?

## CORE CONCEPTS AND TECHNOLOGIES

The proposed LPITutor system is developed on three foundational technologies: LLMs, RAG, and adaptive prompt engineering. Each technology plays an important role in ensuring accurate, contextual, and personalized tutoring experiences.

### Large language models

LPITutor exploits a pre-trained LLM (based on the GPT-3.5 architecture) as the central generative engine responsible for producing natural language explanations. The LLM is not fine-tuned on our dataset; instead, we accessed it using API and explored its output through strategic input modulation using prompt engineering and document grounding. The proposed model exhibits strong capabilities in knowledge synthesis, language fluency, and pedagogical expression to make it relevant to generate human-like instructional content.

### LLM and prompt engineering

Prompt engineering guides LLMs in delivering more accurate and targeted responses. It carefully designs LLM queries in a way that LLM responds to specific educational goals (*Sonkar et al., 2023*). It is challenging to customize the tutoring experience to each student's level of understanding and learning style. Effective prompt engineering combined with RAG ensures that the system dynamically refines the tutoring session based on a learner's responses.

Pre-trained LLMs are able to understand, generate, and adapt to natural language that enables them to deliver personalized learning experiences. Recent advances in GPT-4 and similar architectures offer a rich dialogue interface between the learner and the tutor system (*Fu et al., 2023*). LLMs are the foundation of ITS that performs three major tasks: (*i*) understand complex queries, (*ii*) provide contextually relevant responses, and (*iii*) produce explanations based on a vast *corpus* of information (*Nye, Mee & Core, 2023*). LLMs in educational applications offer significant benefits because they can be easily adapted to various subjects and generate feedback on various learning tasks, ranging from

theoretical explanations to real-time problem solving (*Jiang & Jiang, 2024*). This broad flexibility increases the tutor's ability to meet a variety of learning methods and levels of student comprehension.

The proposed LPITutor employs a dual-layer prompt engineering strategy: (*i*) a static layer that contains pedagogically aligned templates (*e.g.*, "Explain the concept of ¡topic¿ using examples relevant to ¡student profile¿"), and (*ii*) a dynamic layer that injects retrieved content and learner-specific metadata into the prompt structure. This enables the system to adjust the style, depth, and complexity of its responses based on the learner's level of knowledge, query history, and preferred learning style. Prompt variations are automatically selected using an intent classification module and a personalized engine, which ensure that the LLM is oriented towards generating responses with educational relevance and clarity.

## RAG

Apart from harnessing the higher capabilities of LLM, ITS also incorporates RAG to improve its knowledge base and provide more accurate and contextual responses. RAG enables the system to capture critical information from external sources and merge that information into its output to make sure that the response is current and contextual (*Modran et al., 2024*). The method is practical in academic scenarios where accuracy and context are imperative.

Figure 1 illustrates the RAG method that facilitates dynamic content generation and proactively looks for the most pertinent data in real time to provide a high level of relevance and accuracy. In RAG architecture, the system operates in five main steps. (*i*) The user inputs a query and passes it to the retrieval model. (*ii*) In retrieval model, retrieval engine searches for relevant information from external data sources and obtains relevant results in step (*iii*). Afterwards in step (*iv*) the retrieved results combined with the appropriate prompt are passed to a generative model (LLM). In step (*v*), the generative model processes the prompt and generates a final response and returns it to the user as a final result. This hybrid approach efficiently integrates the knowledge from external databases with the generative power of LLMs to deliver precise and context-sensitive responses. This process improves learning outcomes by providing the most relevant and personalized topic-specific explanation (*Chenxi, 2023*).

To address the challenge of hallucination and ensure that the generated content aligns with verified instructional material, LPITutor integrates a RAG mechanism. The system embeds course documents into a vector space using sentence-transformer-based encoders and stores them in a vector database. When a learner submits a query, it is similarly embedded and matched with this vector database using cosine similarity. The retrieved passages on top k form the context window that is fed into the LLM, thus anchoring the generation of authoritative course content and allowing for explainable answers.

Collectively, these core technologies enable LPITutor to act as a context-sensitive, learner-sensitive, and content-based intelligent tutoring system. Their integration ensures

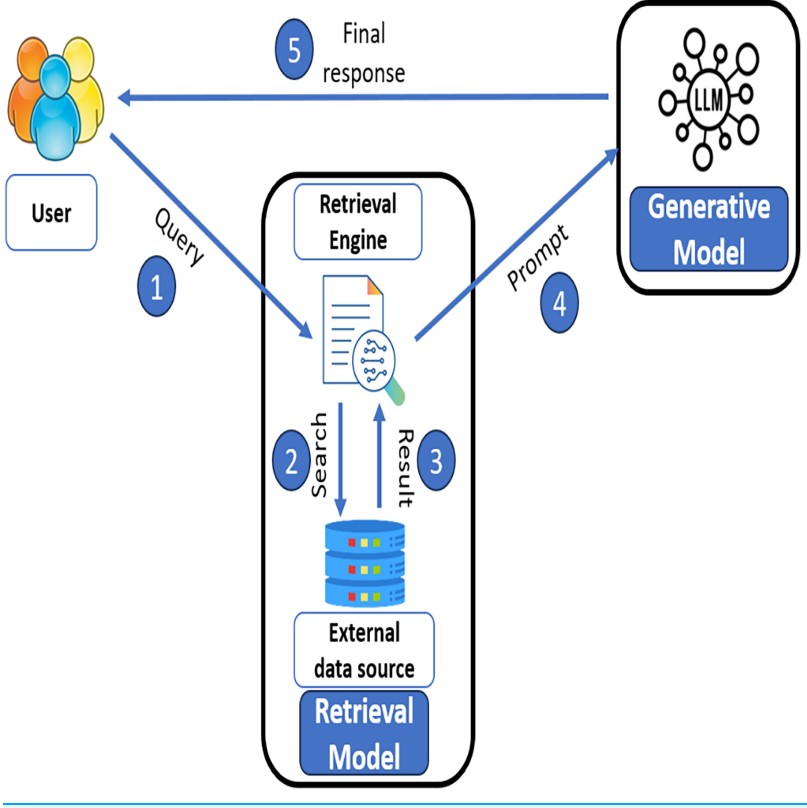

**Figure 1  RAG architecture.**                     

a balance between linguistic fluency, factual accuracy, and instructional personalization, making the system suitable for deployment in formal educational settings.

## LITERATURE SURVEY

LLMs have shown great potential to reshape the educational environment through personalized tutoring systems. These systems can offer personalized educational experiences by leveraging the power of RAG and prompt engineering. Several educational platforms that use comparable architectures that show the integration of LLMs into ITS are gaining popularity. Table 1 explains the state-of-the-art work related to ITS.

*Naya-Forcano et al. (2024)* developed ChatPLT to deliver physics instruction in higher education that shows the adaptability of LLM-powered systems in various domains of study. *Jiang & Jiang (2024)* emphasized the implementation of LLMs in tutoring systems that enable dynamic conversations, which provides a more precise understanding of complex subjects. These models further improve the quality of response by using RAG to retrieve contextually relevant information before generating outputs. This approach optimizes the relevance of the material, as well as the ability of the system to involve learners in active problem solving tasks.

*Chizzola (2024)* highlighted the prompt engineering role in ITS development that is directed to LLM to be more accurate and academically relevant. *Vujinović et al. (2024)* described the importance of the RAG approach in ITS that allows continuous

**Table 1 Literature survey.**

| Ref. | Year | Methodology used | Results | Limitations |
|---|---|---|---|---|
| Naya-Forcano et al. (2024) | 2024 | Development and deployment of an intelligent tutoring system for teaching Physics in higher education using an LLM-based architecture. | Effective at delivering personalized Physics instruction with increased engagement and knowledge retention. | Limited to physics domain, further validation needed across other subjects and educational levels. |
| Jiang & Jiang (2024) | 2024 | Application of LLMs for dynamic tutoring in physics education, leveraging RAG for deep learning and precise understanding. | Enhanced deep learning and student understanding in Physics through interactive and precise tutoring. | Potential issues with bias in LLM responses and a need for more transparent decision-making processes. |
| Chizzola (2024) | 2023 | Development of a generative AI-powered tutoring system with focus on natural language interactions to assist learners. | Showed significant improvement in learner engagement and the ability to tailor tutoring to individual needs. | Still requires further refinement to handle diverse educational subjects beyond initial use cases. |
| Vujinović et al. (2024) | 2024 | Use of ChatGPT for annotating datasets and evaluating its effectiveness as an intelligent tutoring system in education. | Successfully annotated educational datasets, demonstrating LLM's capabilities as a viable tutoring system. | Limited scalability beyond dataset annotation tasks and potential challenges with bias in outputs. |
| Alsafari et al. (2024) | 2024 | Exploration of LLM-powered teaching assistants using intent-based and natural language processing techniques for personalized assistance. | Enabled personalized teaching assistant services, improving learner satisfaction and efficiency. | Lacks detailed strategies for addressing biases in LLM-generated responses and intent recognition. |
| Dou et al. (2024) | 2024 | Design of an LLM-based medication guidance system for providing real-time, personalized recommendations using advanced information retrieval. | Improved accuracy of medication recommendations, showing promise for future applications in educational contexts. | Focused on a specific domain (medication guidance), limiting its generalizability to other educational fields. |
| Wu et al. (2024) | 2024 | Introduction of SecGPT, focusing on secure execution and isolation architecture for LLM-based systems in educational contexts. | Ensured security in LLM-powered environments, safeguarding educational data while maintaining scalability. | Primarily focused on security, lacking detailed educational case studies beyond execution isolation. |
| Schneider, Schenk & Niklaus (2023) | 2023 | Implementation of LLM-based autograding systems for short textual answers in educational assessments. | Improved grading efficiency, particularly in short textual answers, with minimal human intervention. | Limited to autograding, and struggles with more complex responses requiring deeper understanding. |

improvement of learning materials by drawing from the most current available sources. This real-time retrieval capability ensures that students can access the latest knowledge in their field of study, enhancing the credibility and reliability of the tutoring system.

*Alsafari et al. (2024)* discussed that LLM-powered systems are transforming the role of teaching assistants by providing more personalized and intent-based support to learners. *Wu et al. (2024)* supported SecGPT based on an LLM powered environment that not only provides personalized education but also maintains security and user privacy. *Schneider, Schenk & Niklaus (2023)* discussed important points focused on LLM-based classification and assessment where AI-based decisions must be transparent and free from bias. *Dou et al. (2024)* explained the working of ShennongMGS which is LLM-based medication guidance system that successfully integrated domain-specific knowledge to provide highly accurate, context-sensitive information.

*Lee (2024)* presented a computer tutoring system that integrates Generative AI with RAG to tailor instruction delivery and map responses to vetted educational materials. The experiment proved that RAG-augmented systems perform far better than conventional static tutors by dynamically accessing applicable knowledge segments and anchoring LLM-synthesized answers in fact-based sources. The method reduces the risk of hallucinated answers and guarantees pedagogical integrity of the system. By integrating domain-specific course content into vector databases, the tutor could contextualize queries from users and provide extremely relevant instructional feedback, setting a new standard for adaptive learning environments. *Teng et al. (2024)* investigated the application of LLM-based intelligent teaching assistants (ITAs) in flipped classroom learning, where students learn through instructional content before attending in-class sessions. The research focused on how such assistants, driven by generative AI, could turn passive consumption of content into interactive, dialogue-based learning. The authors combined LLMs with formal RAG pipelines to make sure that system responses were not only fluent but also pedagogically sound and based on course-specific documents. Interestingly, their results showed a rise in student engagement, conceptual clarity, and satisfaction when RAG-augmented ITAs were used. Such systems facilitated adaptive scaffolding through personalization of explanation and subsequent queries according to learner profiles and prior query history, thus encouraging richer cognitive processing. The application of RAG as a content-validation layer also boosted the credibility of the system as well as cut down on solely probabilistic generation. *Zhao, Zhou & Li (2024)* proposed Chat2Data, a novel interactive system that combines LLMs with RAG and vector databases for real-time data analysis. While their target application area was not education-centric, the architectural principles of the system—especially its reliance on semantic similarity-based retrieval and grounded generation—are directly applicable to educational use cases. In Chat2Data, queries from users are converted to dense embeddings and searched against a structured vector store with relevant analytical documentation. Retrieved context is subsequently utilized to create grounded prompts for LLM-based answers. This process greatly enhanced the interpretability and factual correctness of produced insights, providing a robust framework that may be tailored to intelligent tutoring systems. By mapping model outputs to external, verifiable content, *Zhao, Zhou & Li (2024)* showed that RAG frameworks can realistically be used as a buffer against misinformation and model drift and thus are extremely relevant to knowledge-critical applications such as education.

Although recent advances have seen the emergence of LLM-based ITS, many models often generate contextually relevant but factually ungrounded responses, lack adaptability to individual learner profiles, and do not offer verifiable source references. LPITutor addresses these limitations through the following novel features:

1. **RAG backbone:** Unlike traditional LLM-ITS that generate responses from latent memory, LPITutor retrieves semantically relevant instructional content from a curated vector database of course-specific materials. This ensures factual grounding and minimizes hallucination.

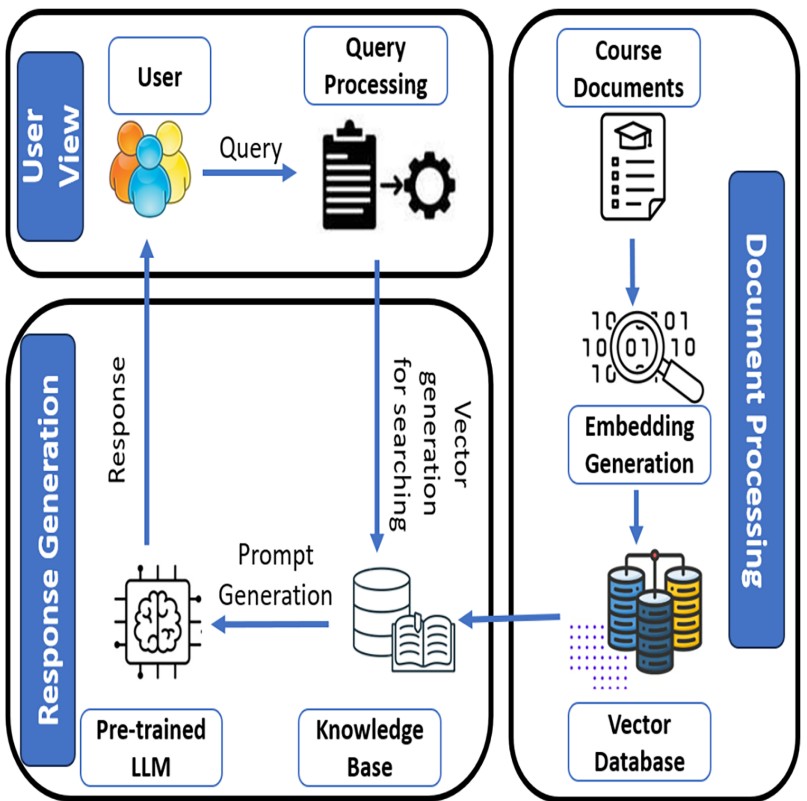

**Figure 2  LPITutor architecture.**               

2. **Learner-aware prompt engineering:** LPITutor constructs dynamic, context-sensitive prompts by incorporating learner profiles, including prior knowledge, query history, and preferred explanation styles. This enables the system to provide adaptive and personalized feedback rather than generic responses.

3. **Traceability and content validation:** Each generated response is supported by traceable document references from the embedded course repository, which supports explainability and allows learners to verify the origin of the content, an essential feature lacking in conventional ITS.

4. **Modular and scalable architecture:** The architecture of the system (Fig. 2) decouples document processing, query processing, and response generation into modular components, allowing LPITutor to be easily extended to new domains or integrated with multiple educational datasets.

These innovations position LPITutor as a hybrid, explainable, and learner-centric ITS, advancing beyond the capabilities of existing black-box LLM tutoring agents.

## LPITUTOR: SYSTEM ARCHITECTURE AND WORKFLOW

ITSs are computerized teaching tools that work like human teachers and provide individualized lessons to many students at once. They depend on AI to make learning more personalized and to do it automatically (*Crow, Luxton-Reilly & Wuensche, 2018*).

ITSs have been used along with more traditional methods, such as cognitive tutors, AutoTutor (*Graesser et al., 2004*), and the ASSISTments tool (*Heffernan & Heffernan, 2014*). But since AI came along, ITSs have been said to be important tools to improve the teaching and learning process, making them a useful addition to modern schools.

In this research, we took a pretrained LLM (GPT-3.5 API) as the pedagogical backbone of LPITutor and did not fine-tune the LLM itself because large-scale model training imposes high computational overhead. Rather, we benefited from its pedagogical strength by an externally modulated prompt generation pipeline that incorporates both query-aware and learner-aware aspects.

Our framework, shown in Fig. 2, is based on the RAG paradigm, in which the user query is initially mapped to a dense representation and matched against a vector database of semantically tagged course documents. The highest ranked documents are fetched and fed into a dynamically formatted prompt that incorporates user profile information, previous interaction context, and pedagogical purpose. This produces extremely personalized responses that remain factually grounded and contextually accurate.

The novelty of LPITutor are:

1. The hybrid prompt engineering approach, which combines static templates with dynamically fetched material and learner-specific metadata.
2. The combination of knowledge-based response generation with domain-specific vector search.
3. The personalization engine, which records student learning styles (*e.g.*, granularity of concepts, style of explanation) to regulate prompt structure and content scope.
4. The adaptive feedback loop, allowing for iterative learning based on learner queries and system confidence scores.

Unlike standard tutoring agents, which depend solely on hardcoded policies or generic LLM responses, LPITutor's pipeline with modularity allows the system to stay scalable, understandable, and sensitive to content as well as learner dynamics. In this section, a brief summary of each module is presented.

## User view

In the user-view phase, the system receives a query from the user. The query can be a question or a request related to the course materials that the user wants to learn. This input query is processed through a query processing system that interprets the user's intent and prepares it for further action. The primary goal of this phase is to understand the requirement of a user using NLP operations. NLP operations are performed for extraction of keywords and other relevant information from the ambiguous and complex queries which are useful for further processing. This phase involves breaking down the query into components that can be more easily mapped to document search or knowledge retrieval tasks.

## Document processing

This phase reads the course materials or relevant documents and is stored in the knowledge base in text vector format. Initially, the relevant documents are collected in the course documents step. These documents will be considered as an external source for this architecture. For further processing, this document is converted into embeddings, *i.e.*, a vectorized form of text that allows us to understand the semantic meaning of the content within the document. These embedding are generated by transformers because it helps to capture the contextual information from the text and makes it easier to match relevant documents according to the user's query. Afterwards, all the generated embeddings are stored in the vector database. This phase is responsible for converting the relevant documents into vectors for an efficient and accurate result.

## Response generator

This phase reads the query from user view and vector database form document processing phase and generates a final response using LLM and appropriate prompt. The vector generated from the query processing step is applied to the knowledge base, and then the appropriate text is retrieved. The extracted text is passed to the LLM with an appropriate prompt for a contextually appropriate response. The response generation process generates the personalized response based on user requirements. This response involves summarizing the text, answering a specific question, or providing an explanation on a specific topic.

LPITutor architecture leverages the vast knowledge and understanding of both general language patterns and domain-specific content to provide precise responses.

## RESULT AND DISCUSSION

LPITutor offers various advantages over general-purpose LLM; (*i*) It provides the customized educational experience to individual students, on the other hand generic LLM generates the generic responses without adapting to student's level, (*ii*) LPITutor has ability to remember previous interaction with student and its level, (*iii*) This architecture build for specific course, that's why it provides the domain specific explanation.

### Experimental design

The effectiveness of a customized approach in real world educational settings is validated through tests that evaluate its retrieval and relevance to actual information (*Verma, 2024*). A prototype was developed that helps generate questions for each topic present in the vector database. Domain experts evaluated the responses and feedback, and the system intentionally provided the wrong answer to determine its functionality. These tests help establish credibility in real-world educational settings. The prototype was tested with domain experts.

The proposed model examines the impact of a structured curriculum on the performance of LLMs in generating educational content. The research uses GPT-4 as a baseline to compare its output under two conditions: (*i*) a pre-defined syllabus, and (*ii*) without syllabus. LLMs process large amounts of information and generate coherent

content across multiple domains. The first condition involves GPT-4 generating content based on an established syllabus, providing clear objectives, topics, and timelines for learning. This ensures that the model generates relevant, aligned content, and maintains logical flow. The second condition involves GPT-4 generating content without a syllabus. This leaves the model independent and rely on its internal knowledge base and contextual understanding of educational content. This model evaluates the effect of a structured syllabus on the caliber, coherence, and applicability of the model's educational content. If a structured syllabus improves performance, it suggests that LLMs can create informative and pedagogically sound educational material. In contrast, if the absence of a syllabus leads to more scattered or less useful content, it needs further refinement or human intervention to ensure alignment with instructional goals.

## Dataset design

The suggested research assesses the model's ability to create learning content and to answer varying levels of complexity. For every subject, the user inputs the questions based on their level of expertise. The questions are grouped as beginner, intermediate and advanced levels of difficulty. The set of answers reflects a progression of learning, starting from fundamental concepts at the beginner level, building upon them at the intermediate level, and culminating in more complex topics at the advanced level. A guided condition is enabled to explore and examine how external guidance influences the model's performance. This state offers a clear syllabus equivalent to the subject, acting as a blueprint to create responses as educational material. The syllabus defines the main elements of the subject, consisting of core principles, key terminologies, and precise learning goals. This mechanism is intended to mimic a truer educational experience where content dissemination is synchronized with pedagogic objectives. The model subsequently creates responses which are directly synchronized with the defined syllabus. It guarantees that the essential ideas are addressed methodically and learning outcomes are achieved. It is a guided process with the aim of utilizing the strength of LLMs like GPT-4, tapping their immense knowledge into organized, targeted answers which are informative and aligned with learning objectives.

The dataset used for evaluation consists of 300 queries distributed evenly across difficulty levels. These questions were either taken from open learning sources or created by subject matter specialists. Every question was manually examined for readability, aptness, and instructional utility. There was a preprocessing step carried out to normalize input form and to ensure linguistic cohesion. For the controlled condition, every question was matched with a syllabus segment related to the subject. It offers a realistic simulation of organized learning environments that is appropriate for assessing both spontaneous and guided generation situations.

By providing this organized dataset and explicit experimental settings, research work guarantees that others can reproduce the setup, baseline performance, and build upon the work using related methods. The dataset contains difficulty label sets of queries and corresponding syllabus content that can be made available for request or released under an

open-access repository for further research into educational AI and language model testing.

## Evaluation metrics

The proposed research used specific metrics to assess the quality and difficulty of GPT-4 answers to various queries. These metrics provided a multidimensional assessment that ensured the accuracy and appropriateness of the content. They allowed a comprehensive evaluation of the model's performance across different difficulty levels and its ability to meet educational standards for different subjects.

Accuracy was assessed by comparing model output with authoritative sources and standard textbooks in the relevant subject to ensure that key facts, definitions, and explanations were aligned with well-established knowledge. This metric was applied to all difficulty levels. The degree to which the model answered the query taking into account the need for in-depth explanations, numerous steps, or the inclusion of several concepts was what determined its completeness. The completeness was used to determine whether the model was capable of appropriately scaling its responses in terms of depth and detail as the difficulty of the questions increased.

The clarity of the model was a determining factor in an educational environment, as it guaranteed that the information was not only accurate but also clearly comprehensible to the learner. The success of the model lay in the efficiency of its capacity to communicate intricate concepts in a format the target group would comprehend. For low-level beginner questions, plain language was adopted, shying away from jargon and rendering straightforward examples. For intermediate-level and challenging questions, more technical terms and detailed explanations were suitable, but the model had to explain concepts logically and coherently. Clarity made the responses correct and easily understandable, enabling students with varying levels of expertise to understand the content. The difficulty alignment measure was utilized to evaluate the model's capacity to deal with questions of different difficulty levels, such that the model's answer was suited to the student's learning stage as intended. GPT-4 could adapt its tone, depth and detail depending on the question's complexity, providing suitable pedagogic answers.

The study tested the performance of the model in answering various questions of various difficulty levels based on different metrics. Coherence was utilized to measure logical information presentation, especially in intermediate and advanced questions where there were several concepts combined. Relevance was utilized to keep the model on topic and directly answer the question without deviating into other information. For higher-level questions, the model produced an answer that was closely related to the question's purpose. By aggregating these measures, the study shed light on the model's strengths and areas of improvement, guaranteeing that the content was of high quality and pedagogical value. The findings indicated that the model performance was effective in leading learners through intricate ideas with a structured and clear explanation.

## Experimental results

While this research demonstrates the potential of the model to generate structured educational content at varying difficulty levels, validating its practical usability requires testing in real-world learning environments. User testing was performed with students to validate the results and assess how effectively AI-generated content supports actual learning outcomes and engagement.

### Accuracy

Measures whether the answer contains the correct information relevant to the question. Accuracy is often calculated as a binary measure, where the model's answer is either correct or incorrect based on expert review.

$$Accuracy = (number of Correct Answer)/Total Number of Answers. \tag{1}$$

### Completeness

Evaluate whether the answer adequately addresses all aspects of the question. Completeness is often evaluated by reviewing whether the key components of the question are answered. No formal equations are used, but expert evaluation can rate completeness on a scale from 1 (incomplete) to 5 (fully complete).

### Clarity

Measures how easy the answer is to understand, especially for the intended audience. Clarity is evaluated on a scale based on language simplicity, sentence structure, and absence of jargon. There is no specific equation for clarity, but reviewers can use a scale from 1 (unclear) to 5 (very clear).

### Difficulty alignment

It checks whether the answer is appropriate for the intended difficulty level of the question (beginner, intermediate, advanced). This is a manual review metric in which reviewers rate the answer based on how well it matches the expected depth and complexity for the given difficulty level (scales of 1 to 5).

### Coherence

Assesses whether the answer is logically organized and flows smoothly. Coherence is evaluated by checking whether the answer maintains a logical structure and progression, such as whether each idea builds upon the previous one. It is typically rated on a scale of 1 to 5 based on expert evaluation.

### Relevance

Measures how closely the answer sticks to the question, avoiding off-topic content. This is usually measured by counting how many parts of the response are directly related to the question. It is often scored on a scale of 1 to 5 (1 = not relevant, 5 = completely relevant).

### Bilingual evaluation understudy

Bilingual evaluation understudy (BLEU) is a machine translation tool that measures the overlap of n grams between a generated sentence and reference sentences. It can indicate

how closely the model response matches expert-written reference answers, especially in phrasing and content expression. The tool compares AI-generated responses to a curated set of expert-written responses for each question. A higher BLEU score indicates better alignment with expert language and structure. However, it may not fully capture semantic correctness or pedagogical relevance in educational settings.

### Recall-oriented understudy for gisting evaluation

Recall-oriented understudy for gisting evaluation (ROUGE) is a set of metrics used for summarization tasks that evaluate recall of n grams, sequences and word pairs from a reference summary. It helps assess how well generated responses capture key points in expert-written answers or provided syllabuses. ROUGE-N and ROUGE-L can be used to assess how well the generated answers reflect the core components and learning outcomes. High ROUGE scores indicate that the model captures the most critical information, aligning with educational objectives. ROUGE is particularly useful in guided conditions where alignment with a structured syllabus is a core requirement.

The study was carried out in a controlled experimental setting that was meant to see how well GPT-4 could create educational content for a range of subject areas and levels of difficulty. Almost 300 graduate-level students and 10 academics from different colleges/universities were included for the experimental purpose. The core goal was to compare the performance of the model under two different conditions—unguided and guided—to determine how the presence of a structured syllabus influences the quality and pedagogical alignment of the generated content. Academic subjects were selected based on their prevalence in standard curricula and diversity in content types. Users with varying levels of subject matter expertise submitted questions.

The study used a model with two types of prompt: Unguided prompt, where the model was given only the question, and Guided prompt, where the model was given the question and subject-specific syllabus as contextual input, to simulate a structured learning environment and align responses with pedagogical goals. All prompts were submitted to GPT-4 using a consistent interface. The responses were evaluated using a rubric based on relevance, depth, clarity, and pedagogical value. They were independently reviewed by two subject experts and disagreements were resolved through discussion or third-party adjudication. The rubric ensured accuracy, completeness, coherence, and support for learning goals.

Based on the evaluation of model performance using the defined metrics, the results for each difficulty level: beginner, intermediate, and advanced were obtained, as shown in Table 2.

### Beginner level results

The model was a highly accurate and complete educational model at the beginner level, providing accurate information on all topics with minimal errors. Its completeness was around 85%, and the responses covered all the core concepts. However, some responses were overly brief, indicating that there is room for more detailed explanations. The model also excelled in clarity, with simple language and clear explanations, making complex

**Table 2 Model performance.**

| Metric | Beginner level | Intermediate level | Advanced level |
|---|---|---|---|
| Accuracy | 97% | 94% | 89% |
| Completeness | 85% | 80% | 75% |
| Clarity | 95% | 88% | 80% |
| Difficulty alignment | 98% | 90% | 85% |
| Coherence | 92% | 85% | 78% |
| Relevance | 96% | 87% | 82% |
| BLEU score | 65% | 52% | 58% |
| ROUGE | 71% | 67% | 61% |

concepts more accessible. The model received a score of 95% for clarity, with minor issues occurring when advanced terms were introduced without sufficient explanation. Overall, the model demonstrated a high level of accuracy, completeness, and clarity at this level.

The model was rated 98% for difficulty alignment at the beginner level, providing answers that matched the simplicity and scope of the questions. It remained logically structured and easy to follow, with a coherence score of 92%. The relevance of the model was 96%, focusing on specific questions without deviating into unnecessary details. However, slight issues arise in abstract or multipart questions, with smoother transitions between points. The coherence of the model was around 92%, with occasional off-topic information in open-ended questions.

### Intermediate level results

The model maintained high accuracy at intermediate level, scoring around 94%. However, it occasionally provided partial answers or misunderstood nuanced questions. The completeness at the intermediate level dropped slightly compared to the beginner level, with 80% indicating that it covered the basics but missed deeper insights or examples. The clarity at the intermediate level remained clear, with an 88% score, but decreased slightly as the questions became more complex. The model used a more technical language suitable for intermediate learners, but could have explained the terms more thoroughly in some cases. Overall, the model remained effective at the intermediate level.

The model effectively addressed the increase in difficulty, with difficulty alignment at 90%. However, some answers were too simple or advanced, causing occasional misalignment. The coherence at the intermediate level was 85%, with logical responses but disjointed information flow on complex topics. The relevance of 87% was slightly increased due to the introduction of extraneous details. The model remained mostly relevant, but some additional information was introduced, making it difficult for learners to follow. In general, the effectiveness of the model in addressing the challenges was commendable.

### Advanced level results

The model accuracy at the advanced level remained strong but slightly decreased to 89%, with problems in specialized and complex topics. Completeness became a significant

**User Query:** "Can you explain the conditions that lead to a deadlock and how it can be prevented?"

**LLM-only Response:** A deadlock occurs when a process waits for a resource that another process holds. There are four conditions: mutual exclusion, hold and wait, no preemption, and circular wait. You can prevent deadlocks by avoiding one of these conditions.

**RAG-enabled LPITutor Response:** Deadlock is a situation in operating systems where a set of processes are blocked because each process is holding a resource and waiting for another. The four Coffman conditions necessary for deadlock to occur are:
- Mutual Exclusion
- Hold and Wait
- No Preemption
Circular Wait
These conditions are explained in detail in Chapter 5 of the OS textbook used in this course. Deadlock prevention can be achieved by eliminating one of these conditions. For example, circular wait can be avoided by imposing a linear ordering of resource types

**Figure 3 Sample Q&A session.**

challenge at this level, with a rating of 75%. The model could generate substantial responses, but sometimes failed to delve into complex topics or miss out on advanced nuances. The clarity at the advanced level decreased to around 80%, with the explanations becoming too dense or technical, making it harder for the learners to grasp the information without additional clarification. Balancing advanced insights with making content understandable was a challenge.

The model scored 85% for alignment of difficulty, and the answers are often too simplistic or complex without an adequate explanation, leading to mismatches between the difficulty of the question and the response. Coherence was also a concern at the advanced level, with complex answers 78% disrupting the logical flow or omitting crucial transitions. The relevance dropped to 82% at the advanced level, and the answers sometimes contained tangential information that was not necessary for the specific question, resulting in unfocused or superfluous details.

Figure 3 showing the real-world behavior of LPITutor using sample question and answer session. This image represents Q&A interactions comparing outputs from the LLM based approach and the LPITutor.

## RAG SYSTEM PERFORMANCE

To evaluate the effectiveness of the RAG component within the LPITutor architecture, we compared three configurations: (i) LLM-based approach without external document

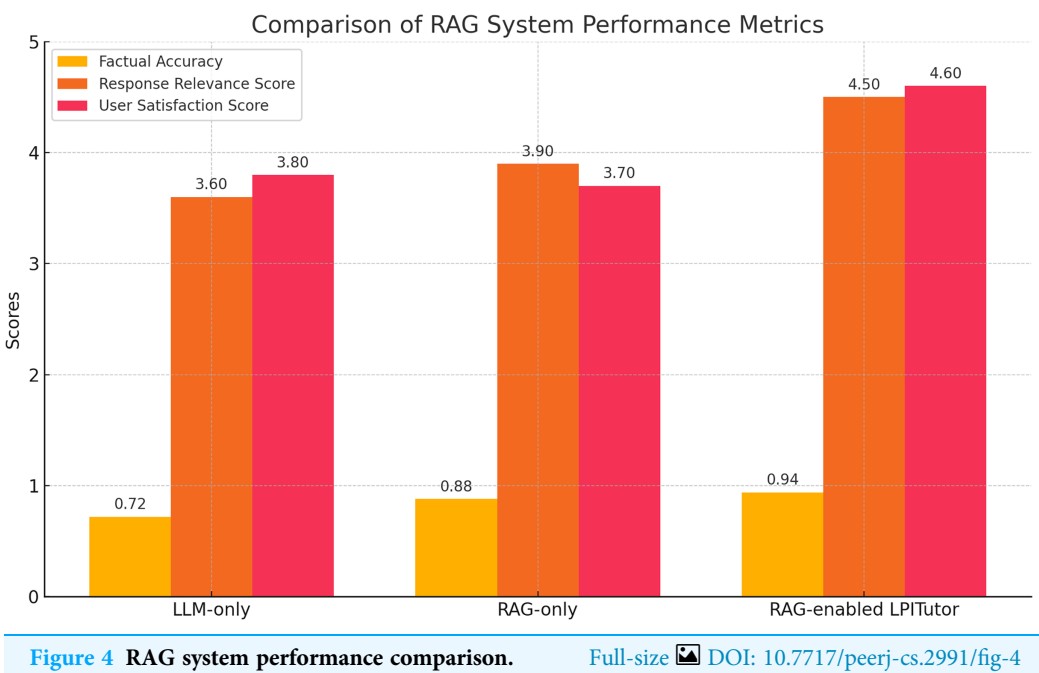

**Figure 4  RAG system performance comparison.**

support, (ii) RAG-based approach, and (iii) LPITutor framework which combines semantic retrieval with dynamic prompt engineering. Each configuration was tested using a fixed set of instructional questions derived from undergraduate computer science lectures. To evaluate performance, we used three metrics: Factual accuracy (FA), which was measured as expert verification of each answer against lecture materials; response relevance score (RRS), based on a 5-point scale rating by independent evaluators; and user satisfaction score (USS), based on the learner's response to clarity.

The experimental results, shown in Fig. 4, clearly highlight the importance of the RAG component. The LLM based approach is fluent but sometimes inaccurate responses with an average FA of 0.72, an RRS of 3.6, and a USS of 3.8. The RAG based approach demonstrated improved factual accuracy (0.88) but was incoherent and non-personalized, resulting in low relevance and user satisfaction scores. However, the LPITutor configuration outperformed both baselines in all metrics with an FA of 0.94, an RRS of 4.5, and a USS of 4.6. These findings support the contribution of RAG to providing grounding for the generative model's output and improving instructional quality.

## ETHICAL CONSIDERATIONS

The deployment of LPITutor focuses on several ethical dimensions. In the following, this section outlines the key mechanisms that this research adopted to ensure ethical compliance and responsible use of AI.

1. **Data privacy and user consent:** LPITutor does not collect personally identifiable information from learners. All query logs used for system evaluation were anonymized and strictly used for research purposes.

2. **Bias and fairness in LLM responses:** LPITutor utilizes a pre-trained LLM; there is always the risk of propagating biases embedded in the model's original training data. To prevent this, we restricted the LLM's generative space by aligning its output to semantically retrieved, domain-relevant content, thereby minimizing hallucinations and off-topic responses. During the system evaluation phase, domain experts reviewed and annotated LLM output for signs of bias, which informed prompt refinement and the development of ethical guardrails for deployment. These combined efforts contribute to promoting fairness, inclusivity, and pedagogical integrity in the responses of LPITutor.

3. **Content verification and feedback loop:** All outputs produced by LPITutor were cross-validated by retrieval-based grounding. In addition, during system evaluation, human-domain experts manually inspected system responses for factual accuracy, tone, and pedagogical correctness. Any suspicious outputs were traced and inspected to assist in future revisions to the prompt engineering pipeline.

4. **Transparency and explainability:** LPITutor provides highlighted source citations for users of the retrieved course materials, allowing them to track the factual source of each response. This is in line with ethical AI standards of accountability and trustworthiness in learning tools.

## CONCLUSION

The results of this study offer valuable insights into the performance of the LLM based personalized intelligent tutoring system using the RAG and prompt engineering models as an educational tool across varying difficulty levels. By analyzing its ability to generate content in response to beginner, intermediate, and advanced questions, several strengths and weaknesses of the model become evident. These findings have significant implications for the application of LLM in educational settings, particularly in areas where content complexity and clarity are crucial. The proposed model is an effective educational tool for novice learners, providing accurate, clear, and relevant information. However, as the questions became more complex, the model performance decreased slightly. At the intermediate level, accuracy decreased to 94%, completeness decreased to 80%, and clarity decreased, indicating difficulty in balancing technical language with increasing difficulty level. At the advanced level, accuracy dropped to 89% and completeness to 75%, indicating that it struggles to handle more specialized and complex subjects. The model's limitations suggest it is not yet ready for stand-alone use, particularly in handling complex and specialized content.

In the future, several areas can be explored to further improve the performance and effectiveness of models, such as the generation of educational content. The model can be improved for advanced learners. In addition, the model can be enhanced to generate questions for multiple subjects in different languages at the same time.

## ACKNOWLEDGEMENTS

The authors acknowledge Ms. Navin Kumar for critically reviewing the technical report and giving constructive suggestions in improving the manuscript quality.

### Funding

The authors received no funding for this work.

### Competing Interests

The authors declare that they have no competing interests.

### Author Contributions

- Zhensheng Liu performed the experiments, authored or reviewed drafts of the article, and approved the final draft.
- Prateek Agrawal conceived and designed the experiments, prepared figures and/or tables, and approved the final draft.
- Saurabh Singhal performed the experiments, performed the computation work, prepared figures and/or tables, and approved the final draft.
- Vishu Madaan performed the experiments, analyzed the data, authored or reviewed drafts of the article, and approved the final draft.
- Mohit Kumar analyzed the data, authored or reviewed drafts of the article, and approved the final draft.
- Pawan Kumar Verma conceived and designed the experiments, analyzed the data, performed the computation work, prepared figures and/or tables, and approved the final draft.

### Data Availability

The raw data is available at Zenodo: Verma, P. K. (2024). LLM based Personalized Intelligent Tutoring System. Zenodo. https://doi.org/10.5281/zenodo.14207595.

The complete source code is available at GitHub and Zenodo:

- https://github.com/phdpawan/LPITutor.

- Verma, P. K. (2025). LLM based Personalized Intelligent Tutoring System. Zenodo. https://doi.org/10.5281/zenodo.15831324.

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
