# Peer review of "LPITutor: an LLM based personalized intelligent tutoring system using RAG and prompt engineering"

_PeerJ Computer Science, doi:10.7717/peerj-cs.2991_

## Round 0.1 · original submission · Major Revisions

You should address all the comments, and in particular a better definition of the research questions, how you trained the LLM and discuss in more details the movelty of the proposed approach.

**Language Note:** The review process has identified that the English language must be improved. PeerJ can provide language editing services - please contact us at [email protected] for pricing (be sure to provide your manuscript number and title). Alternatively, you should make your own arrangements to improve the language quality and provide details in your response letter. – PeerJ Staff

Reviewer 1 ·

Basic reporting

- The paper is written in clear, professional English with no major grammatical issues. The language is technically correct and unambiguous.

- The introduction provides sufficient background on intelligent tutoring systems (ITS) and the role of LLMs in education. Relevant prior literature is appropriately referenced, though some recent studies (e.g., 2024) might need more detailed discussion.

- The authors have shared the code and data on Zenodo, adhering to the journal’s data sharing policy.

Experimental design

- The research question is well-defined, focusing on how LLMs can be used to create a personalized ITS that adapts to different learner levels. The study clearly states how it fills the gap left by traditional rule-based ITS.

- The experimental design appears rigorous, with domain experts evaluating the system’s responses. However, more details on the ethical considerations (e.g., data privacy, bias in LLM responses) would strengthen the paper.

- The methods are described in sufficient detail, particularly the RAG architecture and prompt engineering techniques. However, more information on the dataset used for training and evaluation would improve reproducibility.

- Clarify how LPITutor differs from existing LLM-based ITS in the literature review.

Validity of the findings

- The underlying data are provided, and the results are statistically sound. The evaluation metrics (accuracy, completeness, clarity, etc.) are well-defined and appropriate for the study.

- The conclusions are well-stated and linked to the original research question. The authors acknowledge the limitations of the model, particularly in handling advanced-level queries, and suggest future improvements.

- The study encourages replication, especially in improving the model for advanced learners and generating questions in multiple languages. The rationale for replication is clearly stated, adding value to the literature.

- Discuss potential user testing and feedback from real students to validate usability in practical settings.

- Consider expanding on LPITutor’s adaptability to different educational subjects beyond the tested dataset.

Additional comments

- The paper could benefit from a more detailed discussion of ethical considerations, particularly regarding data privacy and potential biases in LLM-generated content.

- The dataset used for training and evaluation is not described in detail, which could hinder reproducibility.

- The results show a decline in performance at advanced levels, which could be further explored in future work.

·

Basic reporting

-

Experimental design

-

Validity of the findings

-

Additional comments

The paper is generally well-written, but I have a few suggestions for you to consider:

1. for the 'Core Concepts and Technologies Section'

-Consider elaborating further on the details of each core technology, such as LLMs, RAG, and prompt engineering.

-The "LLM and Prompt Engineering" sections and "LLMs in Education" could be merged into one section, as they share overlapping themes.

2. for 'Evaluation Metrics'

- Consider incorporating qualitative metrics, such as BLEU (Bilingual Evaluation Understudy) or ROUGE (Recall-Oriented Understudy for Gisting Evaluation)

- Consider providing a clearer explanation of how each metric is calculated; e.g., use simple examples or illustrations to help readers better understand these metrics.

3. Demonstrating RAG System Performance:

- Consider attaching sample Q&A sessions to showcase the system's performance in real-world scenarios.

Reviewer 3 ·

Basic reporting

The paper contains multiple grammatical errors and typos that should be corrected.

There are several instances of repetition, and the overall flow of the content could be improved.

Additionally, the article would benefit from a more comprehensive background on the research problem, including how the proposed approach differs from similar models.

Several technical aspects require further depth and clarification. For example, the authors should provide details on how the LLM was pre-trained and fine-tuned, which transformer model was used, and what NLP techniques were applied for preprocessing tasks.

Experimental design

The study lacks a clear hypothesis and well-defined research questions.

The case study is not well-defined—there is no mention of the specific audience or educational level for whom the model was used. Furthermore, the evaluation metrics are vaguely presented, with no clear technical explanation of how they were measured. Providing more precise details on these aspects would strengthen the paper.

Validity of the findings

There are other similar methods using RAG and Prompts with LLMs in educational context. The authors didn't discuss the novelty of this work and how it differs from similar research.

Additional comments

Overall, the authors present a promising idea by integrating LLMs and RAG into ITSs. However, the paper lacks sufficient technical depth and novelty. The evaluation of the model's application is not clearly explained, and there are ambiguities regarding the quantified data. Key details are missing, such as how the evaluation metrics were measured, the number of users who interacted with the system, and how many experts participated in assessing the results.

---

## Round 0.2 · accepted · Accept

One reviewer agreed that you improved the paper with respect to the minor comments, so this version is feasible for publication.

·

Basic reporting

I think the revised manuscript meets the publication requirements.
Two good improvements:
-Quantitative Metrics Strengthened: The inclusion of NLP evaluation metrics (BLEU, ROUGE) alongside traditional criteria (accuracy, completeness, etc.) significantly enriches the analysis and boosts confidence in the reported results.

-Clear Innovation Highlighted: The integration of the personalized engine/module with the RAG architecture represents a distinct and valuable contribution to intelligent tutoring systems (ITS), enhancing adaptability.

Two aspects can be improved:
-Formula Presentation can be improved

-Future Student Performance Evaluation Suggested: While the current expert/user evaluation is valid, outlining a concrete plan (e.g., pre/post-test designs, longitudinal studies, specific learning outcome metrics) to evaluate LPITutor's impact on student learning performance and outcomes in future work would significantly strengthen the research roadmap and practical relevance.

Experimental design

No comment

Validity of the findings

No comment

Additional comments

NA